# Alzheimer’s and Parkinson’s Diseases Predict Different COVID-19 Outcomes: A UK Biobank Study

**DOI:** 10.3390/geriatrics6010010

**Published:** 2021-01-26

**Authors:** Yizhou Yu, Marco Travaglio, Rebeka Popovic, Nuno Santos Leal, Luis Miguel Martins

**Affiliations:** MRC Toxicology Unit, University of Cambridge, Cambridge CB2 1QR, UK; yzy21@mrc-tox.cam.ac.uk (Y.Y.); at857@cam.ac.uk (M.T.); rp636@mrc-tox.cam.ac.uk (R.P.); njs76@mrc-tox.cam.ac.uk (N.S.L.)

**Keywords:** COVID-19, Parkinson’s disease, Alzheimer’s disease, SARS-CoV-2

## Abstract

In December 2019, a coronavirus, severe acute respiratory syndrome coronavirus 2 (SARS-CoV-2), began infecting humans, causing a novel disease, coronavirus disease 19 (COVID-19). This was first described in the Wuhan province of the People’s Republic of China. SARS-CoV-2 has spread throughout the world, causing a global pandemic. To date, thousands of cases of COVID-19 have been reported in the United Kingdom, and over 45,000 patients have died. Some progress has been achieved in managing this disease, but the biological determinants of health, in addition to age, that affect SARS-CoV-2 infectivity and mortality are under scrutiny. Recent studies show that several medical conditions, including diabetes and hypertension, increase the risk of COVID-19 and death. The increased vulnerability of elderly individuals and those with comorbidities, together with the prevalence of neurodegenerative diseases with advanced age, led us to investigate the links between neurodegeneration and COVID-19. We analysed the primary health records of 13,338 UK individuals tested for COVID-19 between March and July 2020. We show that a pre-existing diagnosis of Alzheimer’s disease predicts the highest risk of COVID-19 and mortality among elderly individuals. In contrast, Parkinson’s disease patients were found to have a higher risk of SARS-CoV-2 infection but not mortality from COVID-19. We conclude that there are disease-specific differences in COVID-19 susceptibility among patients affected by neurodegenerative disorders.

## 1. Introduction

The rapid emergence of coronavirus disease 19 (COVID-19) has caused over one million deaths worldwide [1]. The clinical features of patients affected by COVID-19 have been extensively explored, but the predisposing factors contributing to increased transmission and clinical severity remain unclear. Social and ecological health determinants such as air pollution [2,3,4,5] have been suggested to increase the risk of infection and exacerbate COVID-19-related illness. In addition, several comorbidities have been proposed to increase COVID-19 mortality rates, including cardiovascular and respiratory pathologies [6,7]. The predominance of these comorbidities in advanced age, combined with the increased incidence of mortality in elderly patients, suggests that age is a major risk factor for COVID-19 mortality [8]. Given the increased incidence of neurodegenerative diseases with ageing, these observations have raised concerns about the vulnerability of patients living with chronic neurological conditions. Neurodegenerative diseases are a heterogeneous group of diseases that are characterised by the progressive loss of neurons of the central and peripheral nervous systems. Alzheimer’s disease (AD) is the most common neurodegenerative disorder and form of dementia worldwide and is characterised by neuronal loss in the hippocampus and cortical areas leading to cognitive decline, memory loss [9,10]. Parkinson’s disease (PD) is the second most common neurodegenerative disorder, and it is characterised by neuronal loss in the *substantia nigra* [11]. In contrast to AD, only a subset of PD patients develop dementia [11]. However, in both diseases, patients experience a gradual worsening of their clinical condition as neuronal loss progresses, ultimately affecting their quality of life. Importantly, patients affected by neurodegenerative disorders often present with multiple age-related comorbidities [12]. These observations have led several studies to investigate the hypothesis that individuals with neurodegenerative diseases exhibit heightened susceptibility to COVID-19.

In one of the largest cohort studies of COVID-19 in Europe, data gathered from 166 hospitals in England, Scotland, and Wales showed that dementia was among the most common comorbidities in the 20,133 patients hospitalised for COVID-19 after adjusting for age and other confounders [13]. Similarly, a single-centre, retrospective, observational study of 627 patients with COVID-19 in northern Italy showed that dementia and its progressive stages were associated with increased mortality [14], consistent with previous findings [15]. While efforts are still ongoing to determine the exact biological basis underlying this association, a recent community-based study revealed that the APOE ε4 genotype, a genetic risk factor for both dementia and AD, is associated with an increased risk of severe COVID-19 [9,10,16]. Taken together, these findings suggest that a dementia diagnosis may represent an important risk factor for mortality in COVID-19 patients. However, it remains unclear whether COVID-19 susceptibility varies across distinct subtypes of dementia, namely, frontotemporal dementia, vascular dementia and AD. In addition, reports on the association between other neurodegenerative disorders and COVID-19 are sparse, hindering both the interpretation and the implementation of related findings into clinical practice. For instance, it remains unclear whether individuals with parkinsonism, and in particular PD, are at increased risk of COVID-19 death compared to the general population. Parkinsonism is an umbrella term that encompasses a number of different neurological disorders grouped together on the basis of several motor symptoms. The majority of patients affected by parkinsonism have a positive diagnosis for PD [17].

As PD-related pathology often leads to respiratory muscle rigidity and an impairment of the cough reflex at advanced stages of the disease [18], individuals suffering from this condition may be at increased risk of COVID-19 mortality [19]. Although data are thus far limited, it is conceivable that PD patients may be at elevated risk of severe respiratory complications following severe acute respiratory syndrome coronavirus 2 (SARS-CoV-2) infection or even unfavourable outcomes. In addition, it has been suggested that other indirect factors of the pandemic, including increased stress, self-isolation, and anxiety, as well as prolonged immobility, may further exacerbate the outcome of PD patients affected by COVID-19 [20,21]. Currently, there is insufficient evidence to show that a pre-existing diagnosis of PD increases the risk of COVID-19 or mortality, and contradictory results have been reported from small samples of COVID-19 patients. A single-centre, case-controlled telephone survey indicated that morbidity and mortality in patients with mild to moderate PD did not differ from those in the general population [22]. Conversely, a later phone survey showed that the COVID-19 prevalence among PD patients in Italy is higher than the national average [23]. These studies provide useful insights into the vulnerability of PD patients, but several methodological inconsistencies hamper their interpretation. For instance, most studies to date include data from only hospitalised or clinically suspected COVID-19 patients. Because a large proportion of SARS-CoV-2 infections are asymptomatic [24], these inclusion criteria likely underscore some degree of misclassification, as hospital admission rates for COVID-19 depend on the prevalence of community testing and admission criteria, which vary between countries.

Using comprehensive clinical data from the UK Biobank, we hypothesised that a pre-existing diagnosis of dementia, AD or PD may be associated with increased risk of COVID-19 and related death. To assess this hypothesis, we analysed baseline (2006–2010) demographic characteristics and pre-existing diagnoses of 13,338 COVID-19 tested volunteers in the UK Biobank. This dataset contains the primary health records of participants who were tested for COVID-19 since the beginning of this pandemic. We report that a pre-existing diagnosis of dementia or AD predicts the largest risk of COVID-19 and mortality. Conversely, PD diagnosis was associated with an increased risk of SARS-CoV-2 infection but not mortality from COVID-19.

## 2. Methods

### 2.1. UK Biobank Data Sources

The UK Biobank comprises health data from over 500,000 community volunteers based in England, Scotland and Wales. Information about the geographical regions, recruitment, and other characteristics has been previously described [25]. Briefly, between 2006 and 2010, adults aged between 40 and 69 years within close proximity to 1 of the 22 UK Biobank recruitment centres were invited to participate. Individuals had extensive demographic, lifestyle, clinical and radiological information collected. Baseline assessments also included a comprehensive series of questionnaires, face-to-face interviews, physical examinations and blood sampling, with linkages to electronic medical records. Clinical data on neurodegenerative disorders and other comorbidities were cross validated by an algorithm from the UK Biobank, which took into consideration the UK Biobank baseline assessment data (verbal interview), linked hospital admissions data, and death register data [26]. Specifically, the diagnoses of neurodegenerative diseases and dementia relied on consensus between primary care and hospital admissions and/or mortality data [27]. This method has been previously validated using a subset of the UK Biobank participants and was shown to have high accuracies of detecting true positives [28]. The linkage method of COVID-19 results to UK Biobank participants has been previously published [25,29]. The full protocol is publicly available, and summary data can be viewed on the UK Biobank website: www.ukbiobank.ac.uk. UK Biobank ethical approval was granted from the North West Multi-Centre Research Ethics Committee. The current analysis was approved under the UK Biobank application #60124. A detailed list of the variables in the present study is presented in Appendix A. We defined hypertension using the criteria of a diastolic blood pressure ≥90 mmHg or systolic blood pressure ≥140 mmHg. Individual-level data were collected from the UK Biobank on 17 August 2020.

### 2.2. Study Design and Exclusion Criteria

We conducted a cohort study using national primary care electronic health record data linked to in-hospital COVID-19 death data (see UK Biobank data sources). Of the 13,338 participants with available COVID-19 data, 1626 tested positive for COVID-19 between 16 March and 26 July 2020, and 11,712 were negative. The majority of samples tested for COVID-19 were derived from combined nose/throat swabs and analysed by real-time polymerase chain reaction (RT-PCR). In intensive care settings, positive cases were identified by a positive test result for SARS-CoV-2 in a hospital setting (i.e., participants whose tests were taken while an inpatient or while attending an emergency department) or death with a primary or contributory cause reported as COVID-19. More information on the testing procedure can be found on the UK Biobank website [30]. During the same period, COVID-19 testing in England was restricted to hospitalised patients with clinical signs of the disease and healthcare workers. In contrast, all UK Biobank participants included in this study were subjected to COVID-19 testing since the beginning of the pandemic. For our models, we defined a positive outcome as either a positive COVID-19 diagnosis or an in-hospital death in COVID-19-positive cases. Risk factors and covariates used for the present analysis were selected on the basis of clinical interest and prior findings. These risk factors and covariates are shown in Appendix A and included dementia, AD, PD, frontotemporal, vascular dementia, cancer, diabetes, high blood pressure, blood group, age, sex, obesity, respiratory difficulties (chronic obstructive pulmonary disease (COPD) and wheezing), forced expiratory volume (FEV), grey and white matter volume, brain volume, white matter hyperintensity and C-reactive protein (CRP) levels. Obesity was defined based on waist-to-hip ratio measurements. The waist-to-hip ratio is determined by dividing waist circumference by hip circumference, meaning that overweight individuals have higher ratios [31]. We grouped smoking status into current, former and never smokers. CRP total levels were normalised according to total protein levels and log-transformed to fit a normal distribution. Other covariates considered as potential upstream risk factors were population density, social deprivation, average household income, education level, housing type, ethnicity, environmental risk within the workplace (chemical, diesel, dusty, smoke), travel to work and the number of people per household. Deprivation was measured using the Townsend Social Deprivation Index (TSDI [32], with higher values indicating higher deprivation), which was derived from the patient’s postcode for a higher degree of precision. Ethnicity was grouped into white or minority ethnicities. The full list of minority groups used for our analysis can be found in Appendix A. For the analysis of the association between PD and COVID-19, we included both incident (those individuals in whom the diagnosis was recorded after their UK Biobank initial assessment visit) and prevalent cases of PD (individuals diagnosed with PD prior to their initial assessment visit). Under all circumstances, PD diagnoses were derived from self-report or linked Hospital Episode Statistics International Classification of Diseases (ICD) codes, as described elsewhere [33].

Our cohort included 157 participants with parkinsonism, of whom 142 were diagnosed with PD. To account for potential differences between parkinsonism and PD in the context of COVID-19-related vulnerability [19], we built 2 separate models—one with individuals diagnosed with parkinsonism and one with PD patients only. Information on all covariates was obtained from primary care records provided by the UK Biobank. The geographical distribution of each subject included in the analysis is shown in Figure 1.

### 2.3. Statistical Analysis

For our analyses of COVID-19 and mortality, we fitted binomial regressions where the response variables were COVID-19 positivity or COVID-19-related death (Figure 1A). We defined COVID-19-related death as an individual who tested positive for COVID-19 and died.

We first took an exploratory approach and included several putative comorbidities presented in the “Study section and exclusion criteria”. More specifically, we omitted several variables, including FEV, COPD, and environmental risk within the workplace, any brain-imaging-derived variable, and travel to work, due to a large number of individuals having missing data. We then applied an iterative variable selection procedure combining unsupervised stepwise forward and stepwise backward regression analyses to select the most suitable predictor or combination of predictors in our models based on the Akaike information criterion. We calculated the odds or risk ratios and their 95% confidence intervals to quantify the effects of the independent variables on the response variables.

These exploratory models showed an association between pre-existing dementia diagnosis and COVID-19 and mortality. We therefore further pursued the link between neurodegenerative diseases and COVID-19 outcome using the same analysis workflow.

To further confirm whether diagnoses of AD or PD were associated with COVID-19 death, we created a subset of the data to include only participants with the selected neurodegenerative disease. We calculated the odds of dying from COVID-19 while accounting for the comorbidities identified in the previous model, which assessed the association between COVID-19 mortality and neurodegenerative diseases.

All models were built using the MASS package [34] in R. The comparison tables were generated using the Stargazer package [35]. The analysis source code, detailed quality checks and all supplementary material are available in GitHub (https://m1gus.github.io/AD_PD_COVID19/). Statistical significance was defined as *p* ≤ 0.05.

## 3. Results

### 3.1. AD and PD Diagnoses Are Associated with an Increase in SARS-CoV-2 Infections in the UK Biobank Cohort

To explore the links between neurodegenerative diseases and COVID-19, we first estimated the risk between COVID-19 and chronic diseases. Chronic diseases often coexist in older adults [36,37,38]. Therefore, we assessed their risk after adjusting for other existing comorbidities (see Figure 1A for the workflow) for an aged and predominantly white cohort in Great Britain (Figure 1B,C). We found that a pre-existing diagnosis of dementia was associated with the largest increase in the likelihood of testing positive for COVID-19 (OR 3.25; 95% CI 2.73–3.87). This was followed by an increased waist-to-hip ratio (OR 3.07; 95% CI 2.26–4.17), a low education level (OR 1.67; 95% CI 1.38–2.02), a higher number of people per household (OR 1.05; 95% CI 1.03–1.07) and an increased TSDI (OR 1.03; 95% CI 1.02–1.04) (Figure 2A and Appendix A). Our findings confirm previous studies showing that dementia predicts one of the highest risks of COVID-19 in elderly individuals [16,39]. Furthermore, consistent with previous observations [40], our primary analysis identified white ethnicity (OR 0.72; 95% CI 0.66–0.77), cancer (OR 0.81; 95% CI 0.74–0.89) and decreasing age (OR 0.97; 95% CI 0.97–0.97) as significant predictors of SARS-CoV-2 infectivity after adjusting for multiple confounding factors (Figure 2A). As recent studies highlighted links between the global burden of dementia and COVID-19 death [39], we next assessed whether the diagnosis of dementia increased the risk of COVID-19 mortality in the UK Biobank participants. Similar to the COVID-19 models, we found that a diagnosis of dementia was associated with the largest risk of mortality from COVID-19 (OR 4.32; 95% CI 3.33–5.60), followed by male sex (OR 1.44; 95% CI 1.20–1.73), increased age (OR 1.09; 95% CI 1.08–1.07), and an increased TSDI (OR 1.07; 95% CI 1.05–1.09) (Figure 2B and Appendix A). Cancer was negatively associated with an increased risk of mortality in our cohort (OR 0.56; 95% CI 0.44–0.72).

Given the prominent role of dementia in COVID-19 diagnoses, we next sought to increase the granularity of our analysis by examining distinct subtypes of dementia. For the analysis of our PD cohort, we observed that only a small subset of patients was diagnosed with dementia (*n* = 10). Given the small sample size, we proceeded to exclude PD individuals with clinically diagnosed dementia from our analysis and included only PD patients without dementia (*n* = 142). Overall, the cumulative incidence of PD and AD diagnosis among COVID-19-positive patients in our cohort was 1.7% and 1.6%, respectively. Our results show that a diagnosis of AD was strongly associated with SARS-CoV-2 infectivity, with AD patients showing the greatest susceptibility to SARS-CoV-2 infectivity compared to individuals without AD (OR 4.15; 95% CI 3.22–5.34). We also found that an increase in the waist-to-hip ratio (OR 3.08; 95% CI 2.27–4.19) or pre-existing vascular dementia (OR 2.51; 95% CI 1.69–3.71) were also positively associated with COVID-19 (Figure 3A and Appendix A). PD diagnosis also emerged as a strong positive predictor of COVID-19 (OR 1.74; 95% CI 1.34–2.27), although the effect was smaller than that for AD diagnosis. Low education level (OR 1.65; 95% CI 1.36–2.0), a higher number of people per household (OR 1.05; 95% CI 1.03–1.07) or an increased TSDI (OR 1.03; 95% CI 1.02–1.04) emerged as significant predictors of a positive COVID-19 diagnosis (Figure 3). Our analysis shows that patients of white ethnicity (OR 0.72; 95% CI 0.66–0.77) or with a pre-existing diagnosis of cancer (OR 0.81; 95% CI 0.72–0.88) were at lower risk of infection in our cohort, while increasing age did not predict an increased risk of infection (OR 0.97; 95% CI 0.97–0.97; Figure 3A). We further investigated whether there was any difference in the risk of testing positive for COVID-19 in participants with parkinsonism (*n* = 157) compared to a subset of patients with PD (*n* = 142). We show that including all participants with parkinsonism led to similar results (OR 1.71; 95% CI 1.37–2.14; Appendix A). We conclude that diagnoses of AD, vascular dementia and PD increased the risk of testing positive for COVID-19 in UK Biobank participants.

### 3.2. AD Patients Are at Higher Risk of COVID-19 Death

Although our results show that patients with AD, PD and vascular dementia are at increased risk of contracting COVID-19, it remains unclear whether the presence of neurodegenerative disorders may exacerbate the risk of mortality in COVID-19 patients. To address this issue, we examined the characteristics and outcomes of all COVID-19 patients in the cohort using a binary multivariable regression model (Figure 3B and Appendix A). In terms of neurodegenerative diseases, a diagnosis of frontotemporal dementia (OR 16.36; 95% CI 5.44–49.15) was associated with the largest risk of COVID-19 death, but this observation suffers from a small sample size (*n* = 6), so we did not explore frontotemporal dementia further in our analysis. We observed that diagnoses of AD (OR 4.17; 95% CI 2.87–6.05) were associated with COVID-19 death but not diagnoses of PD or vascular dementia. In our model, a pre-existing diagnosis of cancer was negatively associated with COVID-19 death (OR 0.63; 95% CI 0.51–0.79), and no significant association was found with diabetes, C-reactive protein levels or ethnicity and COVID-19 death (Figure 3B). We also found that a higher TSDI increased the risk of COVID-19 adverse outcomes (OR 1.07; 95% CI 1.05–1.09), while an increased waist-to-hip ratio (OR 5.83; 95% CI 2.18–15.58) or male sex (OR 1.38; 95% CI 1.16–1.65) were positively associated with COVID-19 death, but this relationship did not reach statistical significance.

We next focused on the role of AD in COVID-19-related deaths. We first built a model that contained only participants with a positive AD diagnosis. Using this model, we found that none of the previously mentioned comorbidities were significant. This observation indicates that participants with AD were at higher risk of dying from COVID-19, independent of age, sex and other comorbidities (Appendix A). We used a similar workflow to further show that a diagnosis of PD was not a predictor of COVID-19-related death. In our model containing only participants with a positive PD diagnosis, we show that age, the TSDI and obesity were significant predictors of COVID-19-related death, indicating that other comorbidities in participants with PD may explain their risk of dying from COVID-19 (Appendix A).

## 4. Discussion

Despite considerable uncertainty in estimates of COVID-19 outcomes, age and comorbid medical conditions are consistently associated with adverse health outcomes in hospitalised COVID-19 patients [15]. The incidence of neurological conditions, including dementia and neurodegenerative diseases, increases with age, and it has recently been proposed that individuals with a pre-existing diagnosis of dementia may be at increased risk of developing COVID-19 [16]. In previous viral outbreaks of respiratory pathogens, including severe acute respiratory syndrome, Middle East respiratory syndrome, and H1N1 influenza, several reports also highlighted the presence of neurological comorbidities in affected patients [41,42]. In the present study, we found that the largest risk factor associated with COVID-19 was a pre-existing diagnosis of dementia associated with AD, vascular dementia or PD. However, while the diagnosis of AD also predicted an increased risk of COVID-19 mortality, our findings suggest that COVID-19 mortality among PD and vascular dementia patients does not differ from that in the general population.

Thus far, several studies have suggested a relationship between COVID-19 and neurodegenerative disorders. An early report [19] this year showed that a dementia diagnosis was associated with the largest increase in the risk of COVID-19 in the UK Biobank based on a smaller cohort of 507 COVID-19-positive patients from England aged 65 and older. Dementia has also been found to increase the risk of in-hospital mortality in a large cross-sectional analysis of 20,133 patients already hospitalised for COVID-19 in the UK [13], a finding later replicated by two cohort, retrospective studies [14,15]. However, these studies mostly focused on hospitalised individuals and did not include data on patients managed in community settings, such as domestic residences. In addition, in line with government guidelines, testing procedures were limited to individuals with COVID-19 symptoms, meaning that available data fail to include the growing number of people who are asymptomatic or are self-isolating at home due to mild symptoms [13]. Using granular data from the UK Biobank, we developed a more robust analysis pipeline since all participants in our dataset received COVID-19 testing. Because a large proportion of SARS-CoV-2 infections are asymptomatic, this screening protocol is more sensitive for the analysis of COVID-19 and mortality rates among people diagnosed with neurodegenerative diseases.

Our results indicate that AD patients are at increased risk of COVID-19. These findings expand a recent analysis of 1091 COVID-19-positive individuals from the UK Biobank. In this study, Zhou and colleagues employed a logistic regression analysis of pre-existing conditions that are overrepresented in patients with COVID-19 and showed that AD was the most significant risk factor for COVID-19, although its association with increased COVID-19 mortality was not investigated [43]. Here, we build on these earlier observations to show that AD is a major risk factor associated with COVID-19 mortality after accounting for a large number of comorbidities. Several features of AD may increase the risk of COVID-19 adverse outcomes. First, the neuropathology of AD could facilitate COVID-19 complications. Increasing evidence from animal studies suggests that amyloid fibrils induce microglial activation and increased activation of the type-1 interferon (IFN) pathway, a crucial component of COVID-19 [44]. Current theories propose that the IFN response in AD may synergise with COVID-19 upon SARS-CoV-2 infection, creating the ‘perfect storm’ of excessive immune responses and thus exacerbating pathology [45]. Supporting the hypothesis of a neurobiological link between AD and COVID-19 mortality, a recent pathological examination of post-mortem tissue from AD patients demonstrated that the protein expression levels of angiotensin-converting enzyme 2 (ACE2), the entry receptor for SARS-CoV-2, were upregulated in the brains of AD patients [46]. This finding raises the hypothesis that higher ACE2 expression may underscore higher viral load in the brains of AD patents, corroborating a potential link between AD neuropathology and COVID-19 mortality [47]. Finally, the social behaviour of patients with dementia and AD must be considered. Cognitive decline may compromise the ability of individuals with AD to follow the recommendations of public health authorities, increasing the likelihood of contagion and the need for carers [48]. Behavioural and psychological symptoms (BPSD) of dementia and AD, such as motor agitation, intrusiveness, or wandering, may further undermine efforts to maintain isolation.

We also found that PD is associated with a heightened risk of SARS-CoV-2 infectivity but not mortality. This is consistent with two recent studies from Italy showing increased COVID-19 mortality rates among PD patients. One group gathered clinical information on 120 community-dwelling PD patients and reported a mortality rate of 20%, a value significantly higher than that of the general population [49]. The second study found that PD patients of older age (>78 years) displayed increased susceptibility to COVID-19 death compared to younger patients [20]. However, both studies used clinically suspected (non-laboratory confirmed) COVID-19 cases, which complicates their interpretation. As noted by the authors, the increased susceptibility to COVID-19 may have resulted in some patients being incorrectly identified as COVID-19-positive, thus leading to a misclassification of COVID-19-related deaths. Moreover, the accuracy of prevalence data might further be hampered by the existence of asymptomatic cases and the lack of population screening campaigns in Italy. Our results support those of a recent case-controlled study that showed that PD was not associated with any apparent risk of morbidity and mortality compared to the general population [22].

Although the biological basis for the higher mortality rate in AD compared to PD patients remains to be elucidated, a recent commentary suggested that PD neuropathology itself might exercise a neuroprotective effect against COVID-19 [50]. For instance, SARS-CoV-2 binds to the ACE2 receptor, which is highly expressed in dopaminergic neurons of the striatum [51]. However, PD-related neuropathology induces significant degeneration of these neurons, raising the hypothesis of reduced neuroinvasion in these patients, as proposed elsewhere [50]. Second, increased neuronal expression of α-synuclein following acute West Nile virus infection suggests that this protein could function as a native antiviral factor within neurons [52]. Finally, a number of PD drugs have been hypothesised to play a therapeutic role in COVID-19. Among these, accumulating evidence shows that amantadine may inhibit viral replication and protect against severe outcomes in PD patients [53]. The proposed mechanism of action involves a disruption of the lysosomal machinery needed for viral replication [54], and there is preliminary evidence of a protective effect against COVID-19 in a small cohort of PD patients, all taking L-DOPA and having tested positive for COVID-19 [55]. In the present study, none of the PD patients receiving amantadine treatment developed severe complications from COVID-19, and only one patient tested positive for SARS-CoV-2 (Appendix A). Although limited by a small sample size, our preliminary analysis is in line with the hypothesis that amantadine may exert a protective effect against both COVID-19 and mortality. Further clinical studies should be conducted to corroborate the therapeutic utility of amantadine for the treatment of COVID-19.

## 5. Limitations

This study presents several caveats. First, our cohort is not representative of the UK population. For instance, the majority of COVID-19-related deaths in the UK took place in care homes [56]. However, only six patients included in our study were reported to live in long-term care facilities, and none of them tested positive for COVID-19. Therefore, our analysis of patients affected by chronic neurodegenerative diseases included only patients living in domestic residences where rapid changes in social behaviour and domestic care may have affected distinct patient groups differently. This feature might have led us to overestimate the effect of chronic neurodegenerative disorders on the risk of COVID-19 and mortality, and further investigation is needed to confirm the generalisability of our results. In addition, it is important to recognise the indirect effects of the current pandemic on AD patients. Because elderly individuals are at increased risk of severe outcomes from COVID-19, government guidelines recommend the isolation of these individuals and limited contact with their family members [57]. However, social activities as well as time spent with other people are generally considered to help prevent cognitive decline in elderly individuals [58]. Therefore, it is plausible that isolation, albeit necessary, may lead to increased stress and cognitive decline among AD patients [59]. In turn, these behavioural difficulties may exacerbate underlying neurodegenerative disorders, contributing to higher rates of hospitalisation and a higher risk of COVID-19 and mortality [59]. Our findings warrant a more rigorous assessment of functional disabilities in patients with AD and PD and their association with COVID-19 mortality. While these effects are challenging to measure in the midst of a pandemic, large-scale retrospective studies will reveal the full range of implications of current isolation measures on AD and COVID-19. Finally, several potentially relevant comorbidities, such as kidney disease, previous myocardial infarction, and stroke, were not included in data collection. Future studies should validate our results by examining the effect of additional comorbidities as well as other factors that we were unable to examine, including access to personal protective equipment, employment, exposure to environmental hazards and the effect of living in care home facilities.

## 6. Conclusions

In conclusion, we report that a pre-existing diagnosis of dementia or AD predicted the largest risk of COVID-19 and mortality. Conversely, PD patients were found to be at heightened risk of SARS-CoV-2 infection but not mortality from COVID-19. Our results support detailed analyses of the biological mechanisms underlying disease-specific vulnerability to SARS-CoV-2 infection among patients with neurodegenerative disorders. Improved knowledge of these factors is critical to develop appropriate strategies to protect clinically vulnerable patients affected by neurodegenerative diseases during this pandemic.

Our results have important implications for disease management and further highlight an important role of disease-specific neuropathology and management in the potential susceptibility to COVID-19.

## Figures and Tables

**Figure 1 geriatrics-06-00010-f001:**
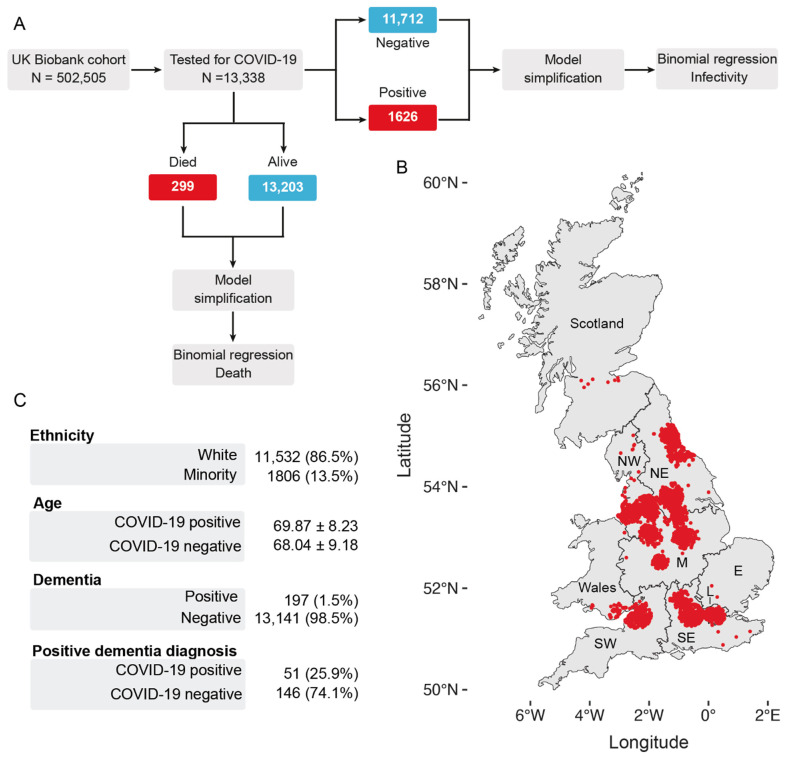
Summary of the workflow and statistics of the analysed cohort. (**A**) Workflow of the analysis. Of 505,505 UK Biobank participants, 13,338 participants were tested for COVID-19 as of 26 July 2020. We modelled the covariates that affect SARS-CoV-2 infectivity or COVID-19-related death using only participants who were tested for COVID-19. We defined a COVID-19-related death as a participant who both tested positive for COVID-19 and died. (**B**) Distribution of the UK Biobank participants in the United Kingdom. (**C**) Descriptive statistics of the cohort analysed. The number of participants in each category precedes their percentage with respect to the total cohort. For the “Age” category, the mean and standard deviation are shown. A full table of summary characteristics is available in Appendix A. SW, southwest; SE, southeast; E, east; M, midlands; NE, northeast; NW, northwest.

**Figure 2 geriatrics-06-00010-f002:**
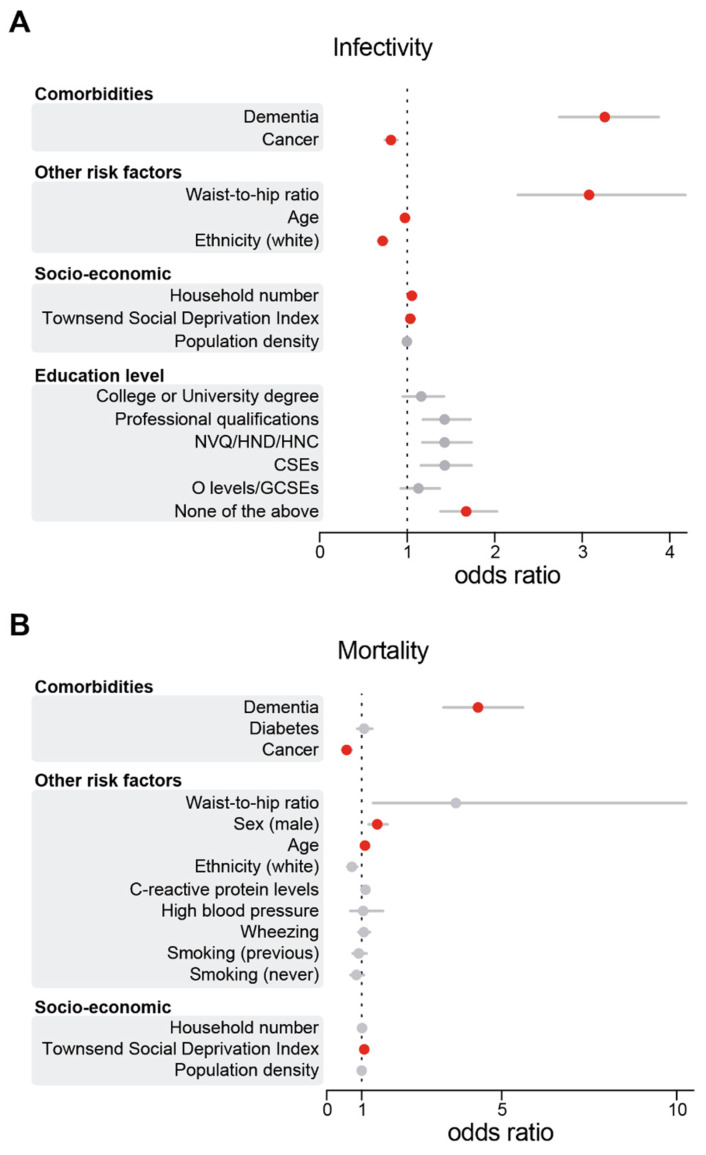
Dementia was the largest risk factor for SARS-CoV-2 infectivity and death. Odds ratios and their respective 95% CIs for the relationship between individual-level characteristics and COVID-19 cases (**A**) or deaths (**B**). For simplicity, characteristics were subgrouped into three categories, namely, ‘comorbidities’, ‘other risk factors’ and ‘socio-economic’. Red indicates significant associations (*p* ≤ 0.05), while grey indicates a lack of significance (*p* > 0.05). The odds ratios for education levels are relative to A-levels. NVQ/HND/HNC, participants who received vocational qualifications such as National Vocational Qualifications (NVQ), Higher National Certificate (HNC) or Higher National Diploma (HND); CSEs, participants with a Certificate of Secondary Education (CSEs); O-levels/GCSEs, participants with either a General Certificate of Secondary Education (GCSE) or a General Certificate of Education (GCE) Ordinary Level (O-levels), a secondary school leaving qualification. This figure is related to Appendix A.

**Figure 3 geriatrics-06-00010-f003:**
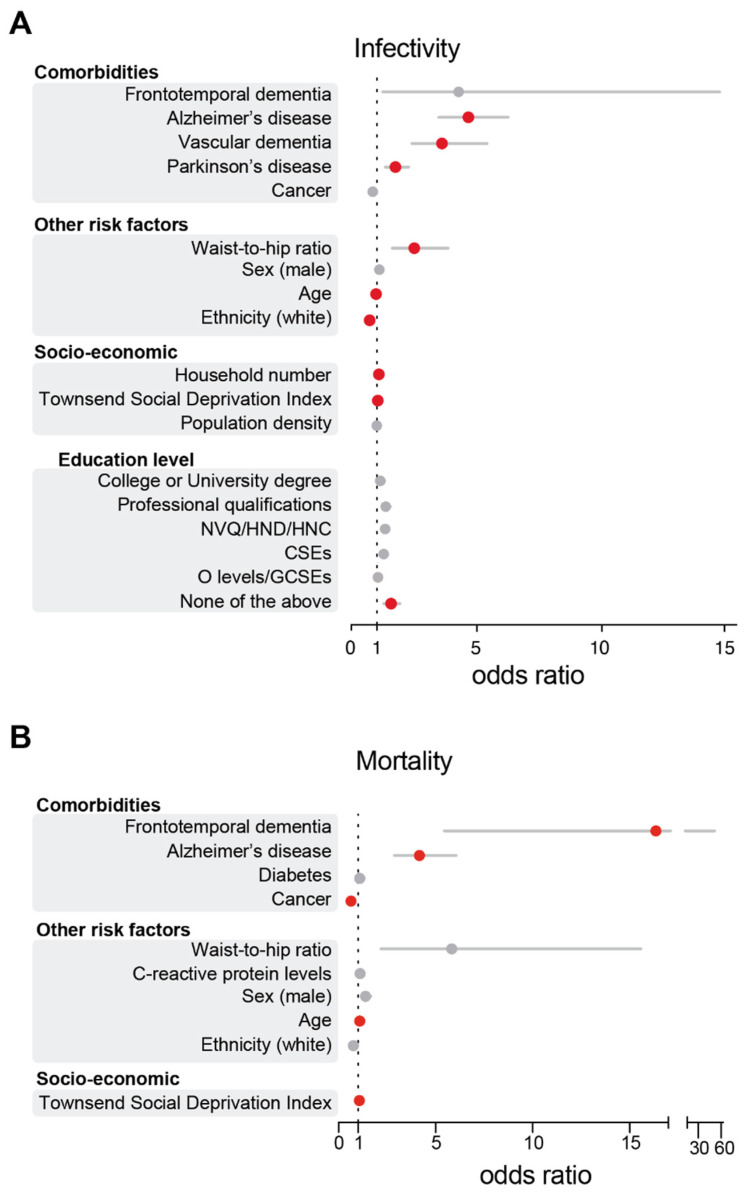
AD diagnosis predicted an increased risk of COVID-19 mortality and infection. Odds ratios and their respective 95% CIs for the relationship between individual-level characteristics and COVID-19 cases (**A**) or deaths (**B**). For simplicity, characteristics were subgrouped into three categories, namely, ‘comorbidities’, ‘other risk factors’ and ‘socio-economic’. Red indicates significant associations (*p* ≤ 0.05), while grey indicates a lack of significance (*p* > 0.05). The odds ratios for education levels are relative to A-levels. NVQ/HND/HNC, participants who received vocational qualifications such as National Vocational Qualifications (NVQ), Higher National Certificate (HNC) or Higher National Diploma (HND); CSEs, participants with a Certificate of Secondary Education (CSEs); O-levels/GCSEs, participants with either a General Certificate of Secondary Education (GCSE) or a General Certificate of Education (GCE) Ordinary Level (O-levels), a secondary school leaving qualification. The numeric values of this figure are shown in Appendix A.

## Data Availability

Data regarding the current analysis can be requested through the UK Biobank and fully reproduced using the code available in the GitHub repository (https://m1gus.github.io/AD_PD_COVID19/).

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
