# Peer review of "Alzheimer’s and Parkinson’s Diseases Predict Different COVID-19 Outcomes: A UK Biobank Study"

_geriatrics, 2021, doi:10.3390/geriatrics6010010_

Round 1

Reviewer 1 Report

The aim of the study is to explore “whether individuals with dementia, AD or PD are at increased risk of COVID-19 infection and mortality using the UK Biobank”.

The major point of interest is the availability of primary health records of 13,338 UK individuals prospectively tested for COVID-19 between March and July 2020; moreover, it seems that the Authors have access to some (not well specified) medical records, thus permitting a classical analysis of risk factors for COVID-19 infection and related death.

Nevertheless, the study is flawed by several major issues (formal, methodological and of content) that make the interpretation of results very problematic and its final format very confusing.

General issues

  • It is clear that the Authors are not familiar with the taxonomy of neurodegenerative disorders. In particular, they mixed the concept of “neurodegenerative” with that of “dementia”. The labels that they use for the primary and secondary analysis are fluctuating and inconsistent. Parkinson’s disease is not a dementia disorder but a movement disorder; only a subgroup of people with Parkinson’s disease can be affected by cognitive impairment in a later stage. Moreover, they do not mention whether Lewy body dementia or parkinsonisms (which include at least PSP, MSA, vascular parkinsonism) were excluded or included under some other labels.
  • Supplementary tables are not accessible for reviewing.

Introduction

  • Many hypotheses are reported as if they were clearly proved. These hypotheses are sustained by mixing opinion (from narrative reviews) with actual evidence, and mixing results from any kind of study design without any distinction (case series, cross-sectional, hospital-based, population-based). The disorders of interest are not clearly defined (see first point above).
  • The Authors take for granted assumptions not completely proved by literature (e.g. “individuals with dementia or neurodegenerative diseases are at higher risk of COVID-19 infection and mortality”).
  • The main result of the study is improperly reported and discussed in the final part of this section.
    • “We found that a pre-existing diagnosis of dementia or AD predicts thelargest risk of COVID-19 infection and mortality. PD was associated with increased risk of infectionbut not mortality from COVID-19. Findings from previous studies suggested an important role for73 dementia in increasing susceptibility to COVID-19 [14]. Here we provide the first systematic analysisof the relationship between COVID-19 and neurodegenerative diseases at the individual level. Ourresults have important implications for disease management and further highlight an important roleof disease-specific neuropathology and management in the potential susceptibility to COVID-19.”

All this part should be deleted.

Methods

  • The following issues are not clear:
    • Which is the source of clinical data/diagnosis
    • According to which procedures and criteria clinical are diagnoses assigned
    • Which were the criteria for testing people for SARS-CoV2 and according to which procedures
  • The statistical analysis should be reported in detail and justified in this section by single hypothesis and not in the Results section.
  • Some statements concerning ethical aspects (whether approval and informed written consent were needed or not; which rights have the authors for accessing data) should be provided.

Results

  • A Table reporting details of the cohort is missing and is needed. This table should provide for the whole cohort and COVID-19 test results crude numbers and frequencies about the basic demographic, clinical (according to reliable categories) and pharmacological data (if available).
  • Looking at the analysis, it seems that “Parkinson’s disease” is included in the group of dementia. This appears to be a mistake: see above “General issues”.
  • In this section general considerations with references are improperly reported. They should be deleted.
  • The logical steps of the multivariable analyses should be more clearly defined.

Discussion

  • Many statements in support of the results are generic and/or not justified, mixing different levels of evidence (experts’ opinion, physiopathological studies, clinical studies), study designs (case series, cross sectional studies) and diagnostic labels (Parkinson’s disease, parkinsonism), without any differentiation: e.g.
    • “A close relationship between COVID-19 and neurological disorders is well established in theliterature.”
    • “We also found that PD is associated with a heightened risk of COVID-19 infectivity but notmortality. This is consistent with a retrospective study conducted in Japan where patients withparkinsonism hospitalised with pneumonia were found to display a lower rate of in-hospitalmortality than age- and sex-matched controls [40].”
    • “Although the biological basis for the higher mortality rate in AD compared to PD patientsremains to be elucidated, it has recently been suggested that PD neuropathology itself might exercisea neuroprotective effect against COVID-19. For instance, SARS-CoV-2 binds to the ACE2 receptor, which is highly expressed in the dopaminergic neurons of the striatum [44]. However, PD-relatedneuropathology induces significant degeneration of these neurons, pointing to reduced neuroinvasion in these patients [17]. Secondly, increased neuronal expression of α-synucleinfollowing acute West Nile virus infection suggests that this protein could function as a native antiviralfactor within neurons [45]. Finally, a number of PD drugs have been hypothesised to play atherapeutic role in COVID-19. Among these, accumulating evidence shows that amantadine mayinhibit viral replication and protect against severe outcomes in PD patients [46]. The proposedmechanism of action involves disruption of the lysosomal machinery needed for viral replication [47]and there is preliminary evidence of a protective effect against COVID-19 in a small cohort of PD patients, all taking L-DOPA and having tested positive for SARS CoV-2 [48].”
  • Literature on Parkinson’s disease and parkinsonism risk should be updated (Risk of Hospitalization and Death for COVID‐19 in People With Parkinson's Disease or Parkinsonism. Mov Disord. https://doi.org/10.1002/mds.28408)

Reviewer 2 Report

Dear Authors 

This paper is very well written and reporting a piece of very sensitive information regarding Alzheimer’s and Parkinson’s diseases predict different COVID-19 outcomes. 

I would like to suggest to add about SARS-CoV-2 and COVID19 spread and its diagnostic importance. See below references will be beneficial. 

a)

Khurshid, Z.; Asiri, F.Y.I.; Al Wadaani, H. Human Saliva: Non-Invasive Fluid for Detecting Novel Coronavirus (2019-nCoV). Int. J. Environ. Res. Public Health 202017, 2225.

Also, improve statistical analysis paragraph. Readability is very difficult. Try to explain in an easy way. 

Please expand limitation and conclusion heading. 

Reviewer 3 Report

In this study the authors say  that a pre-existing diagnosis of Alzheimer’s disease predicts the highest risk of COVID-19 infection and mortality among the elderly. In contrast, Parkinson’s disease patients were found to be at increased risk of infection but not mortality from COVID-19.

In this study the authors say that a pre-existing diagnosis of Alzheimer’s disease predicts the highest risk of COVID-19 infection and mortality among the elderly. In contrast, Parkinson’s disease patients were found to be at increased risk of infection but not mortality from COVID-19.
Overall, the Study provided a meaningful contribution to the field of
research. There are no significant gaps in the cited literature. The results are original and allow incremental advance over prior research results. The research design is appropriate. The right kinds of participants were used. The sample size was adequate. The correct statistics were used. The interpretation of the data
makes sense and logically supports the conclusions. The findings are
important and interesting to the readers. The methods are explained well
enough that the experiments can be replicated. The discussion section
intgrates the findings with relevant theory, rather than simply rehashing the
introduction. The writing is good quality.

Reviewer 4 Report

The paper entitled  Alzheimer’s and Parkinson’s diseases predict different COVID-19 outcomes, a UK Biobank study is very interesting.  The paper needs the following modification.  In the Introduction, please add the hypothesis of the content of the paper and it is essential to justify why this study taken up.

In the result, add percentage of AD patients at risk for death.

Under the subheading of result AD patients are at higher risk of COVID-19 death, add more arguments how you arrive at this information.

Similar argument needs to be added for PD at risk of COVID.

In the conclusion, please add a hypothesis explaining what are the risk factors responsible for COVID risk and death.

Round 2

Reviewer 1 Report

I have no other comments for Authors.

Reviewer 4 Report

The paper has been modified as per reviewers comments. Now suitable for publication.